# Enhancement of Osteoblast Differentiation Using No-Ozone Cold Plasma on Human Periodontal Ligament Cells

**DOI:** 10.3390/biomedicines9111542

**Published:** 2021-10-26

**Authors:** Byul-Bora Choi, Jeong-Hae Choi, Tae-Hyung Kang, Seok-Jun Lee, Gyoo-Cheon Kim

**Affiliations:** 1Corporate Affiliated Research Institute, Feagle Co., Ltd., Yangsan 50614, Korea; cbbrstar@naver.com (B.-B.C.); chlwjdgo@hanmail.net (J.-H.C.); 2Onnuri Pain Clinic, Busan 48585, Korea; tellmekang@naver.com; 3Assemble Plastic Surgery, Seoul 06526, Korea; drlsj69@hanmail.net; 4Department of Oral Anatomy, School of Dentistry, Pusan National University, Yangsan 50612, Korea

**Keywords:** differentiation, no-ozone cold plasma, osteoblast, periodontal ligament cells

## Abstract

Periodontitis is an inflammatory disease that leads to periodontal tissue destruction and bone resorption. Proliferation and differentiation of cells capable of differentiating into osteoblasts is important for reconstructing periodontal tissues destroyed by periodontitis. In this study, the effects of the nozone (no-ozone) cold plasma (NCP) treatment on osteoblastic differentiation in periodontal ligament (PDL) cells were investigated. To test the toxicity of NCP on PDL cells, various NCP treatment methods and durations were tested, and time-dependent cell proliferation was analyzed using a water-soluble tetrazolium salts-1 assay. To determine the effect of NCP on PDL cell differentiation, the cells were provided with osteogenic media immediately after an NCP treatment to induce differentiation; the cells were then analyzed using alkaline phosphatase (ALP) staining, an ALP activity assay, real time PCR, and Alizarin Red S staining. The NCP treatment without toxicity on PDL cells was the condition of 1-min NCP treatment immediately followed by the replacement with fresh media. NCP increased ALP, osteocalcin, osteonectin, and osteopontin expression, as well as mineralization nodule formation. NCP treatment promotes osteoblastic differentiation of PDL cells; therefore, it may be beneficial for treating periodontitis.

## 1. Introduction

Periodontitis is a local inflammatory disease characterized by the loss of periodontal tissues and alveolar bones [1]. If left untreated, periodontal tissue destruction ultimately leads to tooth loss [2]. The fundamental goal of periodontal treatments is to apply reconstruction techniques to restore tissue structure and function. Recent studies have focused on tissue reconstruction or functional recovery after regeneration; of note are the studies on oral mesenchymal stem cells [3,4].

The periodontal ligament (PDL) is a specialized type of connective tissue that connects the surface of tooth roots to alveolar bones [5]. PDL tissue is comprised of a variety of cells—such as endothelial cells, fibroblasts, osteoblasts, and cementoblasts [6]. It also contains undifferentiated mesenchymal cells that can differentiate into mesenchymal cells, which allows them to be harnessed as a source of stem cells [7]. PDL cells are most abundant in the periodontal membrane, where they serve an important role in alveolar bone reconstruction [8]. Thus, osteoblastic differentiation of PDL cells is essential for periodontal tissue reconstruction.

Bone homeostasis involves continuous destruction, resorption, and reformation of bones [9]. During this process, osteoblasts promote bone formation and mediate the differentiation and functionality of osteoclasts through intercellular interactions [10]. Osteoblastic differentiation is critical to bone formation, and each step in this process in which pre-osteoblasts transform into mature osteoblasts requires sequential control through the expression of multiple genes. In the first step, pre-osteoblasts actively proliferate to produce an adequate level of cells for differentiation; this step is regulated primarily through the expression of fibronectin, transforming growth factor-β, osteopontin (OPN), and collagen type 1. Next, the cell cycle is halted and differentiation occurs. During this step, alkaline phosphatase (ALP), bone sialoprotein, and collagen type 1 are expressed, and they induce extracellular substrate maturation. Finally, the extracellular substrate hardens through the accumulation of inorganic minerals such as calcium, and osteocalcin (OC) plays a significant role in the mediation of this substrate mineralization process [11,12,13]. Bone mineralization refers to the deposition of minerals such as calcium in cells or extracellular matrix [14]. Therefore, osteoblast differentiation is a process that ultimately synthesizes bone matrix components and promotes the calcification process [15]. Through the regulation of these steps—including the important step of osteoblastic differentiation—bone reformation can continuously occur, which allows bone homeostasis to be maintained.

Currently, non-thermal plasma is being utilized in a wide variety of medical fields. As non-thermal plasma generates an abundance of electrons, ions, ultraviolet (UV) radiation, hydroxide, and nitric oxide (NO) [16], there are numerous reports of its positive effects on dental tooth bleaching [17], dentin hypersensitivity [18], skin regeneration [19], wound healing [20], bacterial cell death [21], and cancer cell death [22]. Various studies have also actively investigated the use of non-thermal plasma for promoting direct differentiation of stem cells or precursor cells [23,24]; nevertheless, the effect of plasma on PDL cell differentiation remains unclear.

Previously, a technique called nozone (no-ozone) cold plasma (NCP) was developed in which argon gas is used to minimize the generation of ozone and NO compounds. Recently, an NCP device that allows for non-thermal plasma to be used in even more applications has been developed by Feagle Co., Ltd. (Yangsan, Korea); this device is unique, in that it only produces a very small amount of ozone, and it can maintain a temperature of 35 °C or lower [25]. This capability is important for studying the effects of NCP because using this technology in the oral cavity requires the plasma to be created with minimal generation of ozone and NO compounds that may harm the respiratory system.

There is an urgent need to develop a technique to produce plasma that not only promotes PDL cell differentiation but also reduces ozone and NO generation. Therefore, the aim of this study was to determine the effects of NCP on PDL cells that can differentiate into osteoblasts and elucidate the mechanisms involved.

## 2. Materials and Methods

### 2.1. Plasma Device

Figure 1a,b show a schematic diagram and photographs of the non-thermal atmospheric pressure plasma device that was used in this study. The device is composed of a handheld ‘pen’ (handpiece), and the main body consists of a switched mode power supply (SMPS), solenoid valve, gas flow rate controller, main board, and high voltage board. The main body is connected to the handpiece by a cable that is approximately 1 m in length, and the high voltage signal and argon gas are transferred from the main body to the handpiece through the cable. Approximately 3 kV_pp_ of high voltage is transferred from the high voltage circuit inside the main body to the plasma generator in the handpiece, while the argon gas flows at a fixed rate of 1 slm (standard liter per minute) from the gas flow rate controller inside the main body to the plasma generator.

### 2.2. NCP Temperature Measurement

A Fluke 568 contact and infrared temperature gun (CIT; Fluke Corporation, Everett, WA, USA) was used to measure the temperature of the plasma device. The CIT provided accurate temperature measurements over time through a handheld thermocouple that was positioned approximately 10 mm in front of the plasma generating nozzle.

### 2.3. NCP Ozone Concentration Measurement

A Serinus 10 ozone analyzer (OA; Acoem Ecotech, Knoxfield, VIC, Australia) was used to measure the ozone concentration of the plasma device by placing the OA aspirator approximately 10 mm in front of the plasma generating nozzle. The OA uses UV absorption to measure the concentration of ozone over time with an accuracy of 0.001 ppb within a 0–20 ppm range.

### 2.4. Culture and Differentiation of the PDL Cells

The periosteal tissues used in the present study were collected and isolated after receiving informed consent from the patients for their tissue donation and approval from the Ethical Research Committee of the Gyeongsang National University Hospital (GNUHIRB-2009-057). After NCP treatment, osteoblastic differentiation was induced by culturing cells in 24-well plates (3 × 10^4^ cells/well) with osteogenic medium consisting of Dulbecco’s modified Eagle’s medium supplemented with 10% heat inactivated fetal bovine serum, 100 IU/mL penicillin, 100 μg/mL streptomycin, 50 μg/mL L-ascorbic acid 2-phosphate, 10 nM dexamethasone, and 10 mM β-glycerophosphate (Sigma-Aldrich, St. Louis, MO, USA). The cells were allowed to differentiate for 7–21 days, and the media was changed every 2–3 days during the incubation.

### 2.5. Cell Proliferation

The proliferation capability of the cells was assayed using a water-soluble tetrazolium salts (WST)-1 kit (iTSBiO, Seoul, Korea). Briefly, the periosteal-derived cells were seeded into a 24-well plate (3 × 10^4^ cells/well), after which they were exposed to NCP. The PDL cells were divided into different groups and the following four NCP treatment conditions were tested; no NCP treatment (non-treatment; nt), direct application of the NCP onto the PDL cells (direct treatment; Direct), application of a mixture of the NCP with media (plasma-activated media; PAM), and direct application of the NCP onto the PDL cells, with subsequent replacement of the media (direct treatment and media change; MC).

After incubation, the cells were treated with WST-1 (100 μL) and incubated for 2 h at 37 °C. The absorbance of the cells at 450 nm was measured with a microplate reader (Thermo Fisher Scientific, Darmstadt, Germany).

### 2.6. Western Blot Assay

Total protein was isolated from the PDL cells using lysis buffer containing ice cold 150 mM NaCl, 50 mM Tris-HCl (pH 7.4), 1.0% Triton X 100, 0.5 M EDTA (pH 8.0), and a protease inhibitor cocktail (Thermo Scientific, Rockford, IL, USA). The protein samples (25 µg) were separated using sodium dodecyl sulfate-polyacrylamide gel electrophoresis (10% acrylamide), and the separated proteins were subsequently transferred to polyvinylidene fluoride membranes at 300 mA for 2 h. The membranes were then blocked in Tris buffered saline (TBS) with 0.1% Tween 20 containing 5% skim milk at room temperature for 2 h. The membranes were incubated overnight at 4 °C with primary antibodies targeted against beta-catenin (β-catenin) at a dilution of 1:2000, runt-related transcription factor 2 (Runx2) at a dilution of 1:2000, bone morphogenetic protein 2 (BMP2) at a dilution of 1:2000, and glyceraldehyde 3-phosphate dehydrogenase (GAPDH) at a dilution of 1:5000. Subsequently, the membranes were incubated with a horseradish peroxidase conjugated secondary antibody (dilution of 1:5000) for 2 h at room temperature, after which they were imaged using an AI 680 Imager (GE Healthcare Life Sciences, Buckinghamshire, UK).

### 2.7. Alkaline Phosphatase Staining

The PDL cells were fixed with a 3.7% formaldehyde and 90% ethanol solution and gently washed with TBS, after which they were stained with fast 5-bromo-4-chloro-3-indolyl phosphate and nitroblue tetrazolium (BCIP/NBT) alkaline phosphatase substrate (Sigma-Aldrich, St. Louis, MO, USA) for 10 min in the dark at room temperature. The colorimetric reaction was stopped by removing the BCIP/NBT solution and washing the cells twice with distilled water.

### 2.8. Alkaline Phosphatase Activity

The PDL cells in the constructs were washed with PBS and sonicated in an ice-cold bath with 0.1 M Tris buffer containing 0.5% Triton X-100. After the cell lysate was centrifuged, the ALP activity in the supernatant was analyzed using a Lab assay^TM^ ALP kit (Wako Pure Chemicals, Osaka, Japan). The cell lysate was mixed with working assay solution. After incubation at 37 °C for 15 min, stop solution was added to the above mixture to stop the reaction, and the absorbance at 405 nm was measured on a microplate reader). The total protein content of sample aliquots was determined using the Bradford method.

### 2.9. Quantitative Real Time PCR

Total RNA was extracted using TRIzol Reagent (Life Technologies, Eugene, OR, USA). cDNA was then prepared using reverse transcription (One-step PreMix kit; iNtRON Biotechnology Inc., Seongnam-si, Korea) and PCR amplification was performed using a SsoAdvanced universal SYBR Green Supermix (Bio-Rad, Hercules, CA, USA) on a CFX 96 Real Time System (Bio-Rad, Hercules, CA, USA). Relative quantification was achieved using the comparative 2^−ΔΔCt^ method. All the samples were run in triplicate and normalized to a housekeeping gene (GAPDH). The following specific primers were used:Alkaline phosphatase-1 Forward, 5′-TTTGGTGGATACACCCCC-3′Alkaline phosphatase-1 Reverse, 5′-GCCTGGTAGTTGTTGTGAGC-3′Osteocalcin Forward, 5′-AGAGACCCAGGCGCTACCT-3′Osteocalcin Reverse 5′-CTGGGAGGTCAGGGCAAG-3′Osteopontin Forward, 5′-GGTCACTGATTTTCCCACGG-3′Osteopontin Reverse, 5′-GTCCTTCCCACGGCTGTC-3′Osteonectin Forward, 5′-TGGAGGCAGGAGACCACC-3′Osteonectin Reverse, 5′-TCCTTCTGCTTGATGCCG-3′GAPDH Forward 5′-ACTGGCATGGCCTTCCGT-3′GAPDH Reverse 5′-CCACCCTGTTGCTGTAGCC-3′


### 2.10. Alizarin Red S Staining

The PDL cells were washed with PBS and fixed with 10% formaldehyde for 15 min. The fixed cells were then stained with a 2% alizarin red S (pH 4.2) solution (to detect mineralization) for 2 min at room temperature and washed twice with tap water. The cells were then washed in ice cold 70% ethanol in distilled water. The stained cells were photographed and mineralization was detected. The stained cells were then incubated with 100 mM cetylpyridinium chloride (Sigma-Aldrich, St. Louis, MO, USA) for 2 h at room temperature. The absorbance of the cells was then measured at 570 nm using a microplate reader (Thermo Fisher Scientific, Darmstadt, Germany).

### 2.11. Statistical Analysis

All the data are expressed as means ± standard errors (SE), and the data analysis was conducted using IBM SPSS Statistics 20 (SPSS, Chicago, IL, USA). Statistical differences were calculated utilizing a one-way ANOVA. Differences in letters between bars (a, b, c, d, e) indicate statistically significant differences between groups (*p* values < 0.05, one-way ANOVA with Duncan’s post hoc test).

## 3. Results

### 3.1. Measurement of the Gas Temperature and Ozone Concentration of NCP

Since the maximum time for treating NCP to PDL cells was 2 min, temperature and ozone concentration were measured for 15 min. The CIT was used to measure the temperature of the samples at 1, 3, 5, 10, and 15 min. The measured temperatures were 25.4 ± 0.03333, 26.5 ± 0.05774, 28.6 ± 0.05774, 29.3 ± 0.03333, and 29.7 ± 0.12019 °C, respectively (Figure 1c).

To monitor the generation of ozone after switching on the NCP, the ozone concentration was also measured at 1, 3, 5, 10, and 15 min. The measured levels were 0.001 ± 0.00, 0.004 ± 0.00033, 0.005 ± 0.00033, 0.006 ± 0.00, and 0.006 ± 0.00 ppm, respectively (Figure 1d).

### 3.2. Effects of NCP on PDL Cell Proliferation

To examine the effects of the NCP treatment on PDL cell growth, the specific treatment conditions for each of the four groups were administered for 30 s, 1 min, and 2 min. The WST-1 assay was performed on each group after two and four days of the NCP treatments. After two days, the 30 s NCP treatment PAM and 1 min NCP treatment MC groups each showed a slight increase in cellular growth (Figure 2a). After four days, each of the groups with the NCP treatments showed a general decrease in proliferation compared to the groups without the NCP treatment (Figure 2b). Based on these results, the MC group exhibited the minimum inhibitory effect of the NCP on PDL cell growth; hence, this condition was applied in subsequent experiments.

### 3.3. Effects of NCP on Osteogenic Differentiation of PDL Cells

To determine the effects of the NCP on PDL cell osteoblastic differentiation, a Western blot was used to examine the changes in protein expression of β-catenin, Runx2, and BMP2, all of which act as early indicators of osteoblastic differentiation (Figure 3a–c). After the 30 s, 1 min, and 2 min NCP treatments were completed, the PDL cells were supplied with differentiation media to induce differentiation. The results showed that while no significant changes in the expression of β-catenin were observed after six hours of cell proliferation and differentiation, after one day, 30 s of the NCP treatment led to an approximately 1.3-fold increase in β-catenin expression, while the 2 min NCP treatment sample showed an approximately 2.2-fold increase in β-catenin expression. On the third day of differentiation induction after the NCP treatment, the expression of Runx2 and BMP2—which are the two most important proteins that are involved in cell differentiation into osteoblasts—were found to increase with just 30 s of the NCP treatment.

The activity of ALP, a key protein in the bone formation process, was monitored to directly visualize the effects of the NCP on PDL cell differentiation. The ALP activity levels in the untreated PDL cells were first examined using BCIP/NBT staining. As shown in Figure 3d, after one week of the NCP treatment, the ALP activity in the plasma-treated PDL cells was again measured with BCIP/NBT staining. The number of slightly purple stained cells increased in the 30 s and 1 min NCP treatment samples after one week; after two weeks, darker stained cells were observed in the 30 s and 1 min NCP treatment samples. In addition, the optical density (OD) of the ALP activity showed that there was not a significant NCP-induced increase in ALP activity after one week of the NCP treatment; however, after two weeks, the 30 s and 1 min NCP treatment samples exhibited approximately 1.5-fold and 1.4-fold increases, respectively. The 2 min NCP treatment sample did not show a significant change in ALP activity, however (Figure 3e,f).

The effects of the NCP treatment on PDL cell osteoblastic differentiation were once again analyzed based on the real-time PCR results for the mRNA expression of ALP, OC, osteonectin (ON), and osteopontin (Figure 4). After one week, the NCP treatment did not show a significant effect on the expression of the ALP, OC, or ON genes, but the osteopontin mRNA showed an approximate two-fold increase in the 1 min NCP treatment sample. After two weeks, the 30 s and 1 min NCP treatment samples showed ALP gene expression increases of approximately 1.8-fold and 2.7-fold, respectively, while the 1 min NCP treatment sample showed OC, ON, and OPN expression increases of approximately 1.8-fold, 1.7-fold, and 2.3-fold, respectively.

Finally, the facilitating effect of NCP treated-PDL cell differentiation on new bone formation was examined using Alizarin Red S staining and calcium nodule formation in the PDL cells. As shown in Figure 5a, after three weeks of the NCP treatment, the Alizarin Red S staining showed more intensely stained PDL cells in the 1 min and 2 min NCP treatment samples. The OD of each of the Alizarin Red S-stained samples showed that the increase in bone formation was significant for the 1 min and 2 min NCP treatment samples (Figure 5b).

## 4. Discussion

This study was conducted to investigate the effects of a newly developed NCP treatment device on bone regeneration that is central to the treatment of periodontitis and identify the detailed mechanisms. The most noteworthy feature of the NCP that has been developed by Feagle Co., Ltd. is that it produces a negligible amount of ozone during plasma generation, unlike other plasma-generating techniques currently in use. Inhalation of a high level of ozone could potentially cause coughing, throat irritation, or bronchitis (depending on the concentration); in severe cases, pulmonary dysfunction or even lung damage might result [26]. After running the NCP in the current study for nearly 15 min, the level of ozone was 0.006 ppm, which is approximately 10-fold lower than the FDA recommended level of 0.05 ppm, suggesting that long-term use of this instrument in the oral cavity would not have a significant impact on the human body [27].

When the effects of the NCP on PDL cell growth were examined, none of the three NCP treatment conditions showed a significant impact on cell growth. The MC condition was selected for subsequent PDL cell differentiation experiments, as it induced the least amount of cell growth inhibition. Osteoblast and cementoblast PDL cells are capable of differentiation, and these differentiated cells display an ability to form alveolar and cement tissues, respectively [28]. Additionally, bone nodules may form under specific conditions, while bone differentiation proteins, such as ALP and OC, are expressed. The Wnt/β-catenin signaling pathway has also been reported to have a role in PDL cell osteoblastic differentiation [29], as it controls the differentiation of precursor cells that will ultimately proceed to osteoblastic differentiation [30]. Notably, β-catenin—which exhibits a synergistic effect with BMP2—induces bone formation [31], and it binds with the Runx2 promoter to regulate transcription [32]. Therefore, in this study, the way the osteoblast proteins and genes are expressed following the NCP treatment was examined. The results show that the NCP treatment increased the expression of β-catenin in the PDL cells after one day of differentiation, and it increased the expression of Runx2 and BMP2 after three days of differentiation. This suggests that the NCP treatment may stimulate the PDL cells to differentiate into osteoblasts as the expression of osteoblast marker proteins, including β-catenin, Runx2, and BMP2 increases.

ALP, which is the most representative osteoblastic differentiation enzyme marker, is a glycoprotein enzyme that is expressed during the early differentiation of osteoblasts [33]. In this study, when the differentiated cells were stained after one week of the NCP treatment, the 30 s NCP treatment sample showed increased staining intensity, although the ALP activity was not significantly different. Notably, when the differentiated cells were stained after two weeks, both the 30 s and 1 min NCP treatment samples showed increased staining intensity, and the ALP activity was also observed to have increased after the NCP treatment. These results indicate that the NCP treatment promoted the osteoblastic differentiation of the PDL cells. OPN is reported to directly increase ALP activity, as the expression of this highly acidic phosphoprotein increases during the early differentiation of rat calvarial osteoblasts [34]. OC is a non-collagenous protein that is synthesized by osteoblasts, odontoblasts, and cementoblasts, and its expression has primarily been observed during the later stage of mineralization [35]. ON is expressed in a wide array of cells in the bone matrix, and its expression increases with ALP during the second step of osteogenesis, in which matrix secretion via proliferation leads to matrix maturation and, ultimately, matrix mineralization [36]. In this study, the results of real-time PCR showed that, at 1 week after the NCP treatment, NCP treatment induced the expression on OPN mRNA only, but did not affect the expression of the other osteoblastic differentiation marker genes (ALP, OC, ON). However, at two weeks after the NCP treatment-induced differentiation, the cells treated with NCP for 1 min showed the increased the expression of ALP, OC, ON, and OPN. These results reinforce that the NCP treatment may have promoted the osteoblastic differentiation of the PDL cells, and the 1 min NCP treatment (with subsequent replacement of the media) was the most effective NCP treatment condition for facilitating PDL cell differentiation.

Calcium nodule formation is an indicator of normal osteoblast differentiation [37]. In the last experiment of this study, the effects of the NCP treatment on the osteoblastic differentiation of PDL cells were verified using Alizarin Red S staining. The results of this experiment show that the 1 min and 2 min NCP treatment cells exhibited a stronger staining intensity (Figure 5). Thus, collectively, these results suggest that the NCP treatment led the PDL cells to differentiate into osteoblasts by inducing increases in the expression of the proteins and genes that are required for PDL cell osteoblastic differentiation, which ultimately led to an increase in new bone formation.

According to previous studies, plasma treatment of cells leads to facilitated cellular proliferation and differentiation through reactive oxygen species (ROS) and reactive nitrogen species (RNS) that are produced during the plasma generation process [38]. However, these studies applied the plasma treatments to the cells while they were immersed in the media, after which they induced differentiation in the same media. Hence, the cells were exposed for a long time to any ROS or RNS that may have been produced during the plasma generating process. However, in this study, the immediate replacement of the media after the NCP treatment only caused the PDL cells to be exposed to the ROS or RNS dissolved in the media for up to 2 min; therefore, the potential influence of these factors on the NCP-induced PDL cell differentiation is presumed to be low.

## 5. Conclusions

The results of the present study demonstrate that NCP may promote osteoblastic differentiation in PDL cells. Therefore, we believe that NCP is a promising treatment for enhancing alveolar bone formation and reconstruction for periodontitis patients.

## Figures and Tables

**Figure 1 biomedicines-09-01542-f001:**
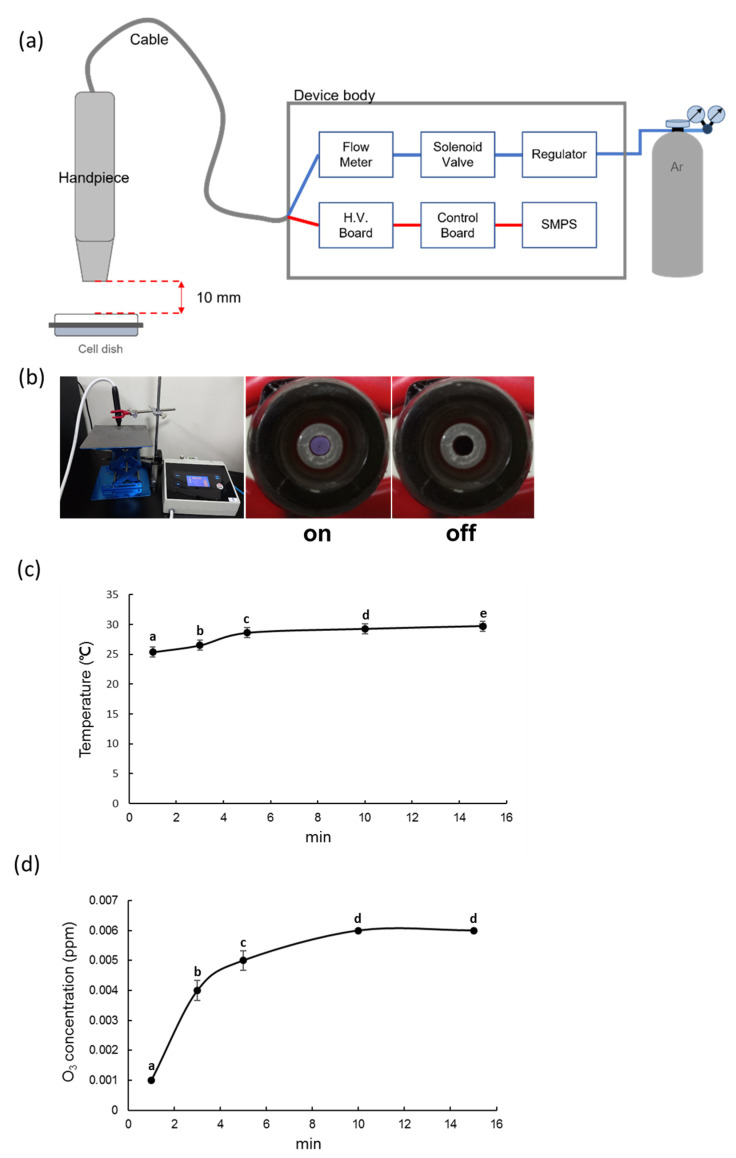
Components of Nozone (no-ozone) Cold Plasma (NCP). (**a**) Experimental setup for an NCP with voltage at 3 kVpp and 1 slm; (**b**) Photograph of the generated NCP; (**c**) Temperature; (**d**) Ozone concentration for NCP treatment. Different letters (a, b, c, d, e) indicate statistically significant differences by one-way ANOVA (*p* < 0.05).

**Figure 2 biomedicines-09-01542-f002:**
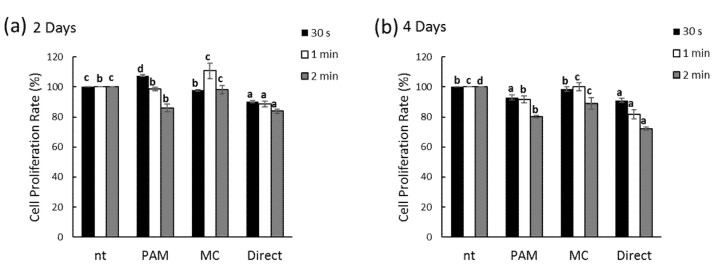
Effect of no ozone cold plasma (NCP) in cell proliferation rate of periodontal ligament (PDL) cells. Cells were treated with various method of NCP at two days (**a**) and four days (**b**). Cell proliferation was determined by water soluble tetrazolium salts (WST)-1 assay. Different letters (a, b, c) indicate statistically significant differences by one-way ANOVA (*p* < 0.05); *n* = 9 for each group. non-treatment; nt, plasma-activated media; PAM, direct treatment and media change; MC, direct treatment; Direct.

**Figure 3 biomedicines-09-01542-f003:**
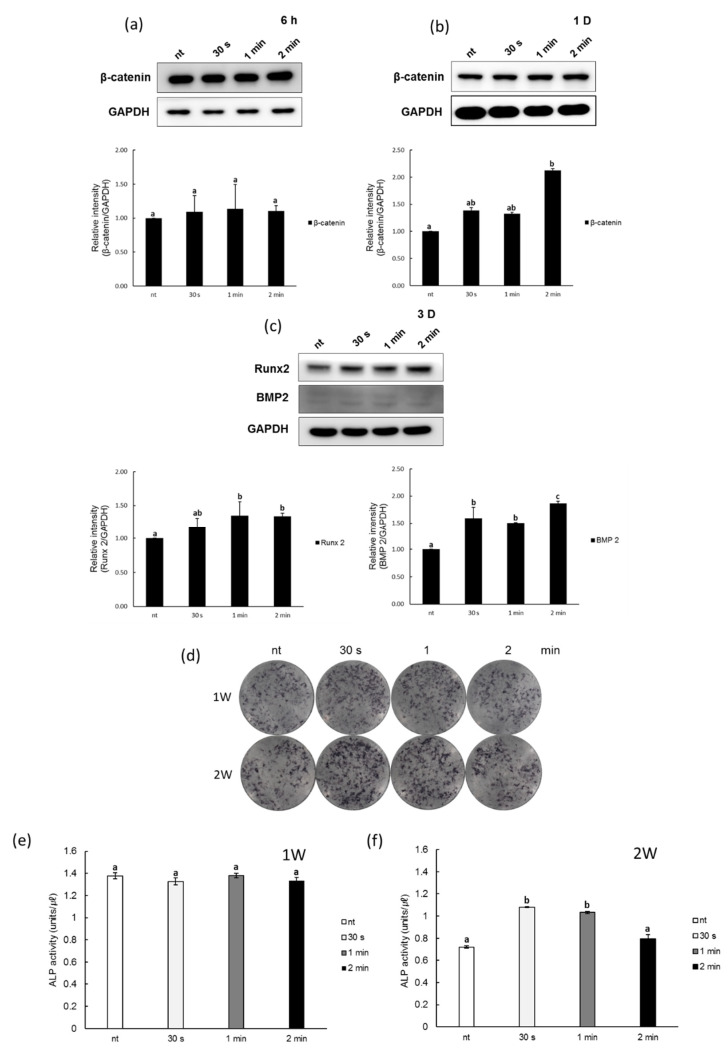
Effects of NCP on osteogenic differentiation in PDL cells. (**a**,**b**) Protein lysates from NCP treated PDL cells were β-catenin, (**c**) Runx2 and BMP 2 were tested by Western blot assay with specific antibodies at various times. After 1 and 2 weeks, (**d**) ALP stain and (**e**,**f**) activity of PDL cells by NCP treatment were evaluated. Different letters (a, b, c) indicate statistically significant differences by one-way ANOVA (*p* < 0.05).

**Figure 4 biomedicines-09-01542-f004:**
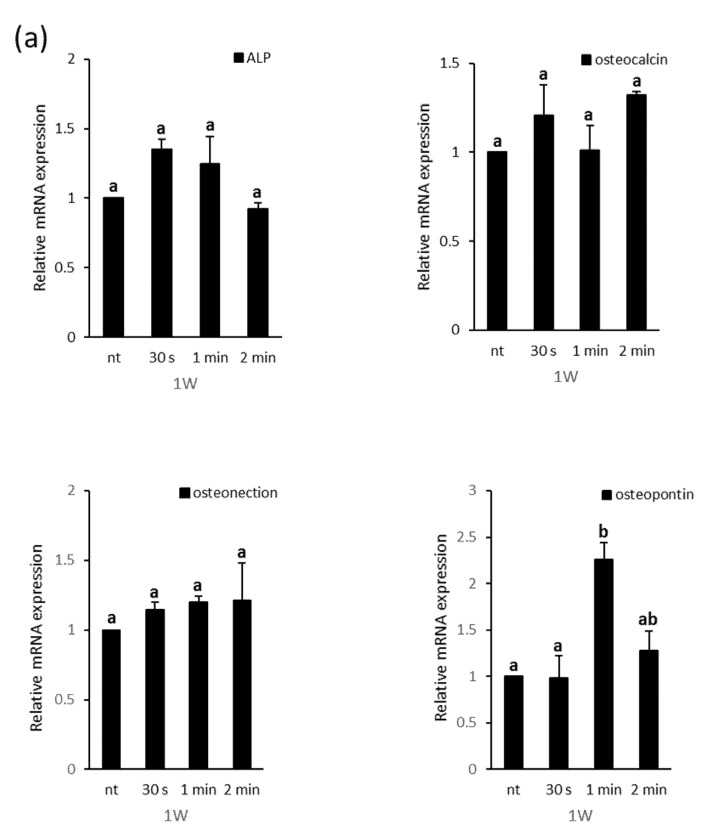
Relative mRNA expression level of the 2^−ΔΔCt^ values by real time PCR, including ALP, osteocalcin, osteonectin, and osteopotin, in PDL cells at (**a**) 1 and (**b**) 2 weeks of osteogenic induction. Different letters (a, b, c) indicate statistically significant differences by one-way ANOVA (*p* < 0.05).

**Figure 5 biomedicines-09-01542-f005:**
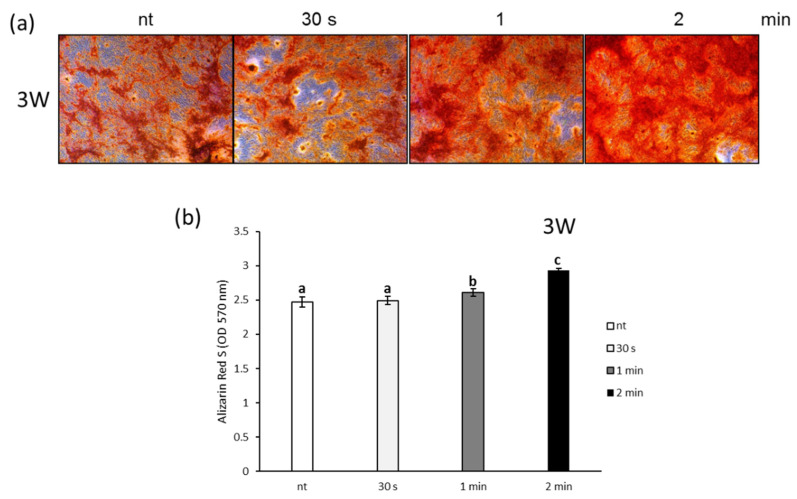
Effects of NCP on calcium deposits in PDL cells. (**a**) After 1 and 2 weeks, alizarin red s stain and (**b**) alizarin red s quantification of PDL cells by NCP treatment were evaluated. Different letters (a, b, c) indicate statistically significant differences by one-way ANOVA (*p* < 0.05).

## Data Availability

The data presented in this study are available upon request from the corresponding author.

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
