# Peer review of "Enhancement of Osteoblast Differentiation Using No-Ozone Cold Plasma on Human Periodontal Ligament Cells"

_biomedicines, 2021, doi:10.3390/biomedicines9111542_

Round 1
Reviewer 1 Report
The loss of tooth causes severe dissatisfaction of quality of life, which is sometimes caused by persistent periodontitis. The induction of osteoblastic differentiation of periodontal ligament (PDL) cells by no-ozone Cold Plasma (NCP) may be beneficial for the treatment of periodontitis. In this manuscript, Choi BBR et al demonstrated that NCP treatment increased ossification in PDL cells. This manuscript is well-organized; however, following points should be clarified.
Major points.
#1. In In figure 2-5, a, b, c, d, are labeled in each bar. What do they mean?
#2. In figure 2d and e, alkali phosphatase (ALP), which is a key enzyme in the bone formation, were not obviously changed in 1W, while ALP was increased in 2W of NCP treatment for 30sec and 1 min. In figure 3, the bone formation that was examined by Alizarin Red S staining was shown to increase in NCP treatment for 2 min. How do authors think?
#3. In figure 4, the expression levels of osteocalcin (OC), osteonectin (ON) and osteopontin (OPN) mRNA were highest in NCP treatment for 1 min. In the section of Discussion, page 11 last sentence and page 12 first sentence, authors claimed that 1 min NCP treatment was the most effective. In figure 2, suppression of cellular growth of PDL cells were induced by 2 min direct irradiation of NCP. Were the PDL cells treated with direct exposure of NCP?
Minor points.
##1. In figure 2, PAM (plasma activated-media) and MC (media change) should be explained in figure legend.
##2. In figure 3a(1D), the band of loading control GAPDH is quite faint. It may be nice to replace it.
Author Response
Comments and Suggestions for Authors Referee
The loss of tooth causes severe dissatisfaction of quality of life, which is sometimes caused by persistent periodontitis. The induction of osteoblastic differentiation of periodontal ligament (PDL) cells by no-ozone Cold Plasma (NCP) may be beneficial for the treatment of periodontitis. In this manuscript, Choi BBR et al demonstrated that NCP treatment increased ossification in PDL cells. This manuscript is well-organized; however, following points should be clarified.
à Thanks for your kind comment.
Major points.
#1. In In figure 2-5, a, b, c, d, are labeled in each bar. What do they mean?
- A post-hoc test is indicated to indicate a significant result for each group. All data were analyzed using one-way ANOVA followed by Duncan’s Significant Difference post hoc test for multiple comparisons.
- Ex) The difference between nt and 1 min was statistically significant, whereas the difference between nt and 30 sec, and the difference between nt and 2 min were not statistically significant. And the difference between 30 sec and 1 min was not statistically significant, whereas the difference between 1 min and 2 min was statistically significant.
- The legend section was modified as below: ☞ Different letters (a, b, c, d, e) indicate statistically significant differences by one-way ANOVA (p < 0.05).
#2. In figure 2d and e, alkali phosphatase (ALP), which is a key enzyme in the bone formation, were not obviously changed in 1W, while ALP was increased in 2W of NCP treatment for 30sec and 1 min. In figure 3, the bone formation that was examined by Alizarin Red S staining was shown to increase in NCP treatment for 2 min. How do authors think?
- As your kind comment, we monitored the fact that the activity of ALP was not affected by NCP at 1 week after the treatment, but the cells treated with NCP for 30 seconds and 1 minute showed the increased ALP activity at 2 weeks after the treatment. Since the time-gap between NCP treatment and increase of ALP activity, we think that this NCP-mediated increase of ALP activity may not be a direct effect of NCP. We think that the direct effect of NCP was restricted to the activation of the proteins for the initial process of osteoblastic differentiation, such as beta-catenin, Runx2 and BMP2 (Figure 3a, b). Furthermore, in these figures, these NCP-mediated increase of these proteins were monitored at the cells treated with NCP for 2 min. As you might know, the increase of ALP activity is one markers for the osteoblastic differentiation. We think that the ALP activity at 2 week after the treatment was increased by 0.5 and 1 min of treatment, but the cells treated with 2 min of NCP did not since 2 min of treatment induced the osteoblastic differentiation faster. The final differentiation of the cells treated with 2 min occurred faster, so that maximum ALP activity might be ended earlier than the cells treated with for 0.5 min and 1 min.
- On the other hand, Alizarin Red S staining is a specific methods for measuring the total accumulation of Calcium content. Therefore, although the increase of ALP activity was not detected in the cells treated with 2 min of NCP at 2 weeks after the treatment, the total calcium concentration at 3 weeks after the treatment can be increased by 2 min of NCP treatment since the final differentiation was ended faster.
#3. In figure 4, the expression levels of osteocalcin (OC), osteonectin (ON) and osteopontin (OPN) mRNA were highest in NCP treatment for 1 min. In the section of Discussion, page 11 last sentence and page 12 first sentence, authors claimed that 1 min NCP treatment was the most effective. In figure 2, suppression of cellular growth of PDL cells were induced by 2 min direct irradiation of NCP. Were the PDL cells treated with direct exposure of NCP?
- As described in our original manuscript, we performed the experiments elucidating the effect of NCP on PDL cell differentiation under the same treatment method (the PDL cells treated with NCP directly, but the media was changed into new fresh media right after the treatment (MC)) based on the result of Figure 2, since this treatment method is not toxic to the cells.
- As we mentioned earlier, the growth of PDL cells is need at the early stage of osteoblastic differentiation. Therefore, we performed our differentiation-related experiments using the NCP methods which shows rare toxicity.
Minor points.
##1. In figure 2, PAM (plasma activated-media) and MC (media change) should be explained in figure legend.
→ Thanks for your kind suggestion. As your comment, we added some sentences explaining the figure legend as below:
☞ Figure 2. Effect of no ozone cold plasma (NCP) in cell proliferation rate of periodontal ligament (PDL) cells. (a) Cells were treated with various method of NCP at 2 (b) and 4 day. Cell proliferation was determined by water soluble tetrazolium salts (WST)-1 assay. Different letters (a, b, c) indicate statistically significant differences by one-way ANOVA (p < 0.05); n = 9 for each group. non-treatment; nt, plasma-activated media; PAM, direct treatment and media change; MC, direct treatment; Direct.
##2. In figure 3a(1D), the band of loading control GAPDH is quite faint. It may be nice to replace it.
- As your kind comment, we have changed it into the figure exposed for the long time of the same membrane.

Reviewer 2 Report
This is an interesting research and the results showed the NCP treatment without toxicity on PDL cells was the condition of 1-min NCP treatment immediately followed by the replacement with fresh media. There are some need to explain about the results in-"our real time PCR results showed that after one week of differentiation induced by the NCP treatment, only the OPN mRNA expres-sion increased after 1 min of the NCP treatment. However, after two weeks of NCP treatment-induced differentiation, the mRNA expression of all three genes (OC, ON, and OPN) increased after 1 min of the NCP treatment."
Major- It is recommended to apply some surface conditions to explain why NCP treatment with low ozone concentration can achieve better cell biological conditions than non-ozone and high ozone surface conditions.
Minor-
introduction- 3rd paragraph “Finally, the extracellular sub-strate hardens through the accumulation of inorganic minerals such as calcium, and oste-ocalcin (OC) plays a significant role in the mediation of this substrate mineralization pro-cess [11-13].” - inorganic minerals such as calcium should be more specific statement.
Results-
- Please pay attention to the measurement accuracy of the instrument and measured significant figures though all the results in the manuscript.
- Can the author provide data on the decay conditions of the ozone concentration on the sample surface over time? Is the sample state under the handheld plasma device uniform, and is there any deviation in the surface area?
- Please explain why not choose the 10 minute plasma surface plateau conditions for comparison? This will greatly support the conclusion.
Author Response
Comments and Suggestions for Authors Referee
- This is an interesting research and the results showed the NCP treatment without toxicity on PDL cells was the condition of 1-min NCP treatment immediately followed by the replacement with fresh media.
- Thanks for your kind comment.
- There are some need to explain about the results in-"our real time PCR results showed that after one week of differentiation induced by the NCP treatment, only the OPN mRNA expression increased after 1 min of the NCP treatment. However, after two weeks of NCP treatment-induced differentiation, the mRNA expression of all three genes (OC, ON, and OPN) increased after 1 min of the NCP treatment."
- As your kind comment, we modified the text in the discussion section as follow:
Page 11, line 340
In this study, the results of real-time PCR showed that at 1 week after the NCP treatment, NCP treatment induced the expression on OPN mRNA only, but not affected on the expression of the other osteoblastic differentiation marker genes (ALP, OC, ON). However, at two weeks after the NCP treatment-induced differentiation, the cells treated with NCP for 1 min showed the increased the expression of ALP, OC, ON and OPN.
- Major- It is recommended to apply some surface conditions to explain why NCP treatment with low ozone concentration can achieve better cell biological conditions than non-ozone and high ozone surface conditions.
- Thanks for your kind suggestion.
- However, we have to explain the NCP (Nozone Cold Plasma) for your better understanding. This NCP generating technology was recently developed by us, which can rarely generate ozone to use the medical plasma in oral cavity, which is directly linked to the respiratory system of our body. As you might know, air ozone level in the work place is restricted since the long-term exposure to high level ozone (above 0.05 ppm) can cause severe damage to our respiratory system. Therefore, to apply the beneficial effect of plasma on oral cavity, ozone issue must be addressed. For this reason, we used NCP which ejects extremely low ozone (below 0.01 ppm) at every surface conditions (since the plasma plum of this device forms at argon gas nozzle, and it is not directly contact with ambient air or target surface). Please consider the fact that we used NCP in this study for safety issue, not for the functional issue.
- Minor-
Introduction
- 3rdparagraph “Finally, the extracellular sub-strate hardens through the accumulation of inorganic minerals such as calcium, and osteocalcin (OC) plays a significant role in the mediation of this substrate mineralization process [11-13].” - inorganic minerals such as calcium should be more specific statement.
- As your comment, we modified our manuscript by adding relevant content as below:
Bone mineralization refers to the deposition of minerals such as calcium in cells or extracellular matrix. (Osteoblastic lysosome plays a central role in mineralization, Sci Adv. 2019 Jul 3;5(7)) Therefore, osteoblast differentiation is a process that ultimately synthesizes bone matrix components and promotes the calcification process. (Bone morphogenetic protein and retinoic acid signaling cooperate to induce osteoblast differentiation of preadipocytes, J Cell Biol. 2002 Oct 14;159(1):135-46)
Results-
- Please pay attention to the measurement accuracy of the instrument and measured significant figures though all the results in the manuscript.
- Thanks for your kind suggestion. As your comment, we checked our measurement data again for several times, and we confirmed that there were no accuracy issue.
- Can the author provide data on the decay conditions of the ozone concentration on the sample surface over time?
- Thanks for your kind suggestion. Unfortunately, we could not get the decay conditions of ozone concentration from this NCP device, since the ozone level at the plasma ejecting end was extremely low. As our data in figure 1d shows, the maximum ozone level of this NCP device is below 0.005 ppm. Please consider the fact that we placed gas inlet of ozone detector at right in front of the end of plasma ejecting nozzle of the device.
- Is the sample state under the handheld plasma device uniform, and is there any deviation in the surface area?
- As our figure 1a shows, during NCP treatment procedures, the distance between the end of handpiece of the device and its target cells, or the sensors of temperature, or the gas inlet of ozone detector was maintained for 1 cm, by fixing handpiece as below:
- Since the distance of the NCP device and its treating surface was restricted for 1 cm and the all sample targets were placed at same plate, we can conclude that the surface condition of sample was unifromed.
- Please explain why not choose the 10 minute plasma surface plateau conditions for comparison? This will greatly support the conclusion.
- Thanks for your kind suggestion.
- Unfortunately, we did not choose the 10 minute of NCP treatment condition, based on our results of Figure 2. Generally, the cells which initiate the differentiation shows the active proliferation before the initiating differentiation process. In our figure 2, however, the treatment of NCP for 2 min was enough for decreasing the cell viability. If NCP was treated on PDL cells for 10 minutes, the severe cell toxicity is expected, and the differentiation stimulating effect might be vanished.
- Please consider the fact that plasma surface plateau conditions can be different from the best plasma treatment condition for the biological effect.

Reviewer 3 Report
There are no line numbers, so it is difficult to comment specifically. Please see the attachment for details.

Author Response
Comments and Suggestions for Authors Referee: 3
There are no line numbers, so it is difficult to comment specifically. Please see the attachment for details.
→ As your comment, we checked the attached file and modified it accordingly. Thank you for checking the modified file.
Round 2
Reviewer 1 Report
Authors inserted 'Different letters (a, b, c, d, e) indicate statistically significant differences by one-way ANOVA (p < 0.05)' in the end of all figure legends. What do a, b, c, d and e mean? Authors described statistic analysis at 2.11. section in page 5 and 6; however, a, b, c, d and e are not explained. Please correct these sentences.
Author Response
Comments and Suggestions for Authors: 1
Authors inserted 'Different letters (a, b, c, d, e) indicate statistically significant differences by one-way ANOVA (p < 0.05)' in the end of all figure legends. What do a, b, c, d and e mean? Authors described statistic analysis at 2.11. section in page 5 and 6; however, a, b, c, d and e are not explained. Please correct these sentences.
Thanks for your kind suggestion. As your comment, the method section was modified as below:
☞ All the data are expressed as means ± standard errors (SE), and the data analysis was conducted using IBM SPSS Statistics 20 (SPSS, Chicago, IL, USA). Statistical differences were calculated utilizing a one-way ANOVA. Differences in letters between bars (a, b, c, d, e) indicate statistically significant differences between groups (P values <0.05, one-way ANOVA with Duncan's post hoc test).
Reviewer 2 Report
The author successfully revised the manuscript based on the suggestions to a large extent, so it can be publicly accepted.
Author Response
Comments and Suggestions for Authors: 2
The author successfully revised the manuscript based on the suggestions to a large extent, so it can be publicly accepted.
☞ Thank you for your kind answer.
Reviewer 3 Report
Now the manuscript is much better than the previous version. Thank you very much for all your efforts.
However, I have the following mandatory suggestions:
1. Please mention the dilution factors for all primary and secondary antibodies in the material and method section.
2. Please depart the Figure 3A 6h from 1D. Make two for them as they were done in a different membrane. Together they are perfect to show. Both time points are different. Better show them as Figure 3A, 3B.
And please include a graphical abstract if possible.
Author Response
Comments and Suggestions for Authors: 3
Now the manuscript is much better than the previous version. Thank you very much for all your efforts.
However, I have the following mandatory suggestions:
- Please mention the dilution factors for all primary and secondary antibodies in the material and method section.
Thanks for your kind suggestion. As your comment, we added some sentences explaining the method section as below:
☞ Total protein was isolated from the PDL cells using lysis buffer containing ice cold 150 mM NaCl, 50 mM Tris-HCl (pH 7.4), 1.0% Triton X 100, 0.5 M EDTA (pH 8.0), and a protease inhibitor cocktail (Thermo Scientific, Rockford, IL, USA). The protein samples (25 µg) were separated using sodium dodecyl sulfate-polyacrylamide gel electrophoresis (10% acrylamide), and the separated proteins were subsequently transferred to polyvinylidene fluoride membranes at 300 mA for 2 h. The membranes were then blocked in Tris buffered saline (TBS) with 0.1% Tween 20 containing 5% skim milk at room temperature for 2 h. The membranes were incubated overnight at 4°C with primary antibodies targeted against beta-catenin (β-catenin) at a dilution of 1:2000, runt-related transcription factor 2 (Runx2) at a dilution of 1:2000, bone morphogenetic protein 2 (BMP2) at a dilution of 1:2000, and glyceraldehyde 3-phosphate dehydrogenase (GAPDH) at a dilution of 1:5000. Subsequently, the membranes were incubated with a horseradish peroxidase conjugated secondary antibody (dilution of 1:5000) for 2 h at room temperature, after which they were imaged using an AI 680 Imager (GE Healthcare Life Sciences, Buckinghamshire, UK).
- Please depart the Figure 3A 6h from 1D. Make two for them as they were done in a different membrane. Together they are perfect to show. Both time points are different. Better show them as Figure 3A, 3B.
As your comment, we separated 6 h and 1 D in figure 3A as below:
And please include a graphical abstract if possible.
Thanks for your kind suggestion. As your comment, we attached a graphical abstract.
